# Micro-mechanical fingerprints of the rat bladder change in actinic cystitis and tumor presence

Laura Martinez-Vidal [1,2], M. Chighizola[3], M. Berardi [4,5], E. Alchera[1], I. Locatelli[1], F. Pederzoli[1,2], C. Venegoni[1], R. Lucianò[6], P. Milani[3], K. Bielawski[4], A. Salonia[1,2], A. Podestà [3✉] & M. Alfano [1✉]

Tissue mechanics determines tissue homeostasis, disease development and progression. Bladder strongly relies on its mechanical properties to perform its physiological function, but these are poorly unveiled under normal and pathological conditions. Here we characterize the mechanical fingerprints at the micro-scale level of the three tissue layers which compose the healthy bladder wall, and identify modifications associated with the onset and progression of pathological conditions (i.e., actinic cystitis and bladder cancer). We use two indentation-based instruments (an Atomic Force Microscope and a nanoindenter) and compare the micromechanical maps with a comprehensive histological analysis. We find that the healthy bladder wall is a mechanically inhomogeneous tissue, with a gradient of increasing stiffness from the urothelium to the lamina propria, which gradually decreases when reaching the muscle outer layer. Stiffening in fibrotic tissues correlate with increased deposition of dense extracellular matrix in the lamina propria. An increase in tissue compliance is observed before the onset and invasion of the tumor. By providing high resolution micromechanical investigation of each tissue layer of the bladder, we depict the intrinsic mechanical heterogeneity of the layers of a healthy bladder as compared with the mechanical properties alterations associated with either actinic cystitis or bladder tumor.

[1] Division of Experimental Oncology/Unit of Urology, IRCCS Ospedale San Raffaele, Milan 20132, Italy. [2] Università Vita-Salute San Raffaele, Via Olgettina, 60, Milan 20132, Italy. [3] C.I.Ma.I.Na and Dipartimento di Fisica "Aldo Pontremoli", Università degli Studi di Milano, Milan 20133, Italy. [4] Optics11, Amsterdam, The Netherlands. [5] LaserLab, Department of Physics and Astronomy, VU University, Amsterdam, The Netherlands. [6] Pathology Unit, IRCCS Ospedale San Raffaele, Milan 20132, Italy. ✉email: alessandro.podesta@unimi.it; alfano.massimo@hsr.it

Elasticity and viscosity characterize the mechanical properties of soft tissues, and play a critical role in defining cell and tissue functions, as well as tissue development, disease progression, and tissue homeostasis[1–3]. Each specific organ has particular mechanical properties, which are altered when homeostasis is disrupted and diseases develop, as it happens for ageing processes, cancer, fibrosis, cardiovascular diseases, and diabetes[4–8].

The bladder is a hollow organ that has to adapt and stretch during filling and voiding, whose functions are achieved through cycles of mechanical relaxation and contractility. Bladder mechanical properties have been reported at macroscale level[9,10], including the characterization of the different macroscopic areas[11]. Alteration of the mechanical properties of the bladder result in a dysfunction of its physiological role, as for many benign bladder pathologies that progress from the formation of stiffer matrix to a more compliant structure[12]. Stiffening has been reported for malignant bladder diseases, which emerged to be associated with high content of collagen fibers in the ECM;[13] moreover, further stiffening of the bladder was reported for patients with relapsing tumor[14]. While these studies on clinical specimens are very informative, they show macroscale measurements and a snapshot of the clinical situation.

This study has focused on two pathological conditions of the bladder: actinic cystitis and urothelial bladder cancer. Actinic cystitis is a pathological condition, which may be caused as a sequelae of pelvic radiotherapy, commonly used to treat prostate and rectum cancer[15,16]. Actinic cystitis is caused by the accumulation of ECM proteins due to chronic inflammation, resulting in scarring and tissue thickening and potentially in end-stage organ failure, with clinically relevant impact toward patients[4].

Bladder cancer is the ninth most common cancer worldwide[17]. It mostly originates from the urothelium, and according to the invasion of the different tissue layers is classified as non-muscle invasive bladder cancer (NMIBC) and muscle-invasive bladder cancer (MIBC). According to the TNM classification system[18], NMIBC are subdivided into pTa and in situ carcinoma (Tis) when the tumor is present in the urotelium, and pT1 when it invades the lamina propria. Conversely, MIBC are segregated into pT2 when the tumor reaches the muscle layer, from which it can further migrate invading the perivesical tissues (stage pT3) and the contiguous organs (stage pT4), thus including prostate, uterus and others[19]. The ability of cancer cells to invade the adjacent tissues is a hallmark of cancer, and in the case of bladder cancer the invasion of the lamina propria[20] or the muscle layer[21] determines a different management work-up. Bladder cancer cells from the urothelium invade the tissue as isolated single cells, cords of cells in single file pattern or small nests[22]. Most of patients with NMIBC experience relapse of the tumor, and therefore undergo multiple cycles of intervention represented by trans-urethral resection of bladder tumor (TURBT) and intra-vesical adjuvant therapy[20]. Eventually, if the tumor progresses and invades also the muscle layer, patients may be candidate to radical cystectomy or multimodal aggressive treatments to attempt and spare the organ[21].

Clinical imaging techniques can detect established fibrosis and initial tumor stages, but there is the clinical need to identify even earlier disease stages and to develop a more tailored follow-up of the disease, in order to identify very initial signs of any disease relapse.

Understanding the link between micromechanical features and clinical phenotypes in bladder malignancies may contribute to improve prognostic classification[6]. Moreover, this knowledge can contribute to the development of novel therapies based on the modification of the micromechanical drivers of carcinogenesis[6]. In addition, a deep understanding of the mechanical features of a healthy bladder may be crucial for any bladder reconstruction purpose, with the goal of developing artificial and mechanically compliant bladders[15,23].

Motivated by the long-term perspective of exploiting the mechanical phenotyping of cells and tissues as an early diagnostic tool for the onset and the progression of diseases like urothelial bladder cancer, we performed force vs indentation measurements to characterize the Young's modulus (YM) of elasticity of bladder tissues, spatially resolved at the microscale. As identification of mechanical markers for diseases must rely on simplified protocols and instrumentation to be performed in the clinics, our strategy was to start with a widely used technique in mechanobiology, that is the atomic force microscopy (AFM) as a reference; and then to move towards the use of a nanoindenter, which is a more handy and potentially easily transferable to the clinics indentation device for soft matter.

Therefore, current project aimed to i) characterize the static elastic properties of the three tissue layers composing the healthy bladder wall at the micro-scale level; ii) identify modifications associated with the onset and progression of pathological conditions (i.e., actinic cystitis and bladder cancer); and, iii) associate such mechanical fingerprints to the gold standard of diagnosis (i.e., histological examination by experienced uro-pathologist).

## Results

**The healthy bladder exhibits regional differences in tissue elasticity.** Murine bladder elasticity was measured by two microindentation instruments: AFM and nanoindenter. Bladder wall emerged to be mechanically inhomogeneous, with local YM covering values ranging from few kPa to hundreds of kPa (Fig. 1a, b). To identify the origin of this mechanical inhomogeneity, we investigated the existence of an association between the YM distribution and the different anatomical layers of the bladder wall. When performing AFM-based indentation focalized on the single tissue layers (Fig. 1a) it was observed that the urothelium had a median YM value of 16 kPa, the lamina propria of 63 kPa and the muscle of 72 kPa (4.20, 4.80 and 4.86 respectively in $Log_{10}(YM/Pa)$). In spatially resolved micromechanical maps acquired using the nanoindenter, we observed that the healthy bladder wall was characterized by a tissue stiffness gradient (Fig. 1b): the urothelium exhibited the lowest YM (10 kPa), which gradually increased when reaching the second bladder tissue layer, the lamina propria (100 kPa), and transiently decreased when reaching the outer muscle tissue layer (70 kPa) (Fig. 1b). Both AFM and nanoindenter instruments provided comparable YM values for the different bladder tissue layers, both in physiology (Fig. 1c) and pathology (for a complete comparison of every physiological and pathological condition and three tissue layers between nanoindenter and AFM, the reader is referred to the Supplementary Fig. 1). Therefore, the following results derive from data collected in duplicate using both AFM and nanoindenter.

**The healthy bladder exhibits temporal evolution in tissue elasticity (stiffening) across aging of the adult animal.** In order to provide an accurate comparison of the bladder mechanics during disease progression in the rat model versus healthy animals, we here characterized the evolution of healthy bladder YM throughout aging process of the animals during their adult life (i.e., from 4 months old, which equals 2 months post-treatment, to 8 months old, that corresponds to 6 months post-treatment). Aging of the animals among their adult life resulted in an overall stiffening of the bladder tissue (Fig. 1d). Of interest, focusing on each tissue layer such a tendency was confirmed. Indeed, the urothelium exhibited a very wide distribution of values, which

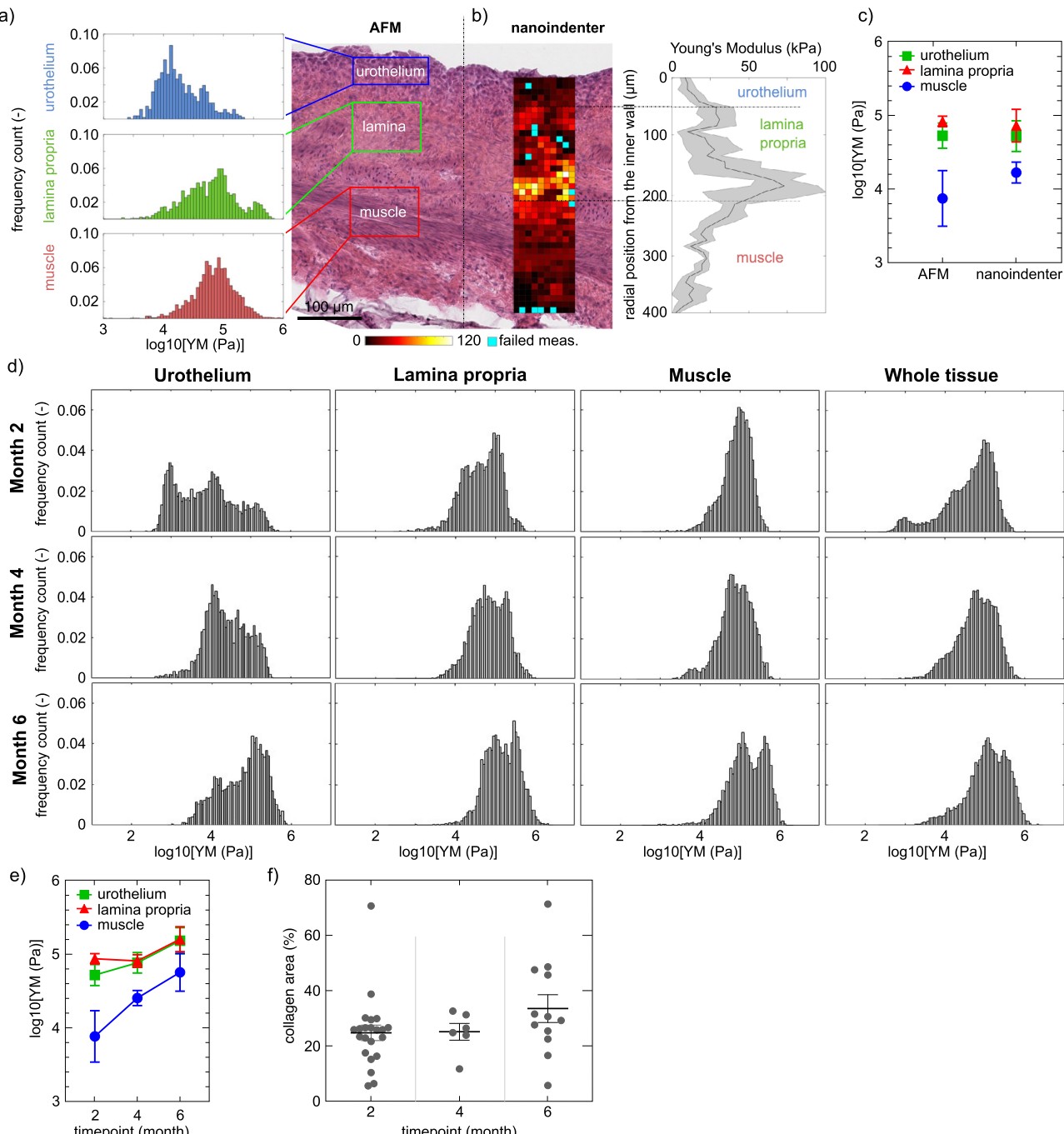

**Fig. 1 Micromechanical map of healthy bladder wall and aging effect on stiffness. a** YM by AFM represented in logarithm in base 10 (log10) in Pa for the different bladder tissue layers. **b** YM values obtained with the nanoindenter. The mechanical heterogeneity of the bladder correlates with its anatomical distribution: there is a gradient of YMs, being the urothelium the softest layer, with increasing stiffness when reaching the lamina propria, and then decreasing over the muscle layers. The association of stiffness gradients with the anatomical bladder tissue layers is shown by overlay of the mechanical map with hematoxylin-eosin staining. Blue pixels indicate rejected measurements. **c** Mean of the median values of a healthy rat bladder wall ± SEM, extracted by the two instruments. AFM and nanoindenter provide comparable results (2-way Anova test showed no statistically significant differences). For a complete comparison of between nanoindenter and AFM, the reader is referred to the Supplementary Fig. 4. **d** Evolution of the stiffness of the different bladder tissue layers (urothelium, lamina propria and muscle) with aging of the adult animal (AFM and nanoindenter data pulled together). The distribution of YM values for the whole tissue (i.e. all layers together) is also shown. $N = 3$ rats per time point, each tissue layer of each single rat is characterized by over 3000 force curve measurements. Data shown are logarithm in base 10 of YM's (in Pa). **e** Mean of the median values of each rat ± SEM. Nonparametric Mann-Whitney test showed no statistical significant differences when comparing the different tissue layers at the studied time points. **f** Quantification of the bladder area that expresses collagen from control rats at different time points; each symbol represents the measurement in a tissue slice, with multiple slices measured for each bladder, in three animals. Source data behind graphs and charts can be found in https://figshare.com/projects/Micro-mechanical_fingerprints_of_the_rat_bladder_change_in_actinic_cystitis_and_tumor_presence/158222.

shifted towards higher values with aging: at Month 2 we could appreciate that the urothelium exhibited a bimodal distribution, which moved towards higher values at Month 6, and the stiffer population got enriched. It could be observed that the highest peak of the urothelium distributions overlapped with the peak of the lamina propria, indicating that the boundaries between the different tissue layers are divided by stiffness transitions, rather than drastic and well-separated tissue boundaries.

Similarly, the lamina propria followed the same tendency: at Month 2 the mean of the median values was $4.72 \pm 0.2$ Log$_{10}$(YM/Pa), up to $5.18 \pm 0.25$ Log$_{10}$(YM/Pa) at Month 6 (Fig. 1e). Likewise, YM increased with aging also for the muscle tissue, and at Month 6 it exhibited a bimodal distribution. Such increase of stiffness did not correlate with an increase of collagen within the bladder tissue (Fig. 1f).

### Microscale mechanics in the actinic cystitis model and during disease progression.
X-ray-irradiated animals developing actinic cystitis were used as model to characterize the mechanical fingerprints of fibrotic bladder tissue (Fig. 2a). X-ray irradiation resulted in an increase of frequency in micturition and a decrease of the urine volume per each micturition (Supplementary Fig. 2). Such biological effect was accompanied by elasticity alterations: irradiation-induced actinic cystitis as a consequence of bladder fibrosis and a more dense ECM deposition, which on the one hand resulted in an increase of bladder wall stiffness, but on the other the elasticity gradient within the bladder wall was maintained (urothelium was characterized by lowest YM, which increased over lamina propria and further decreased when reaching the outer muscle layer) (Fig. 2b).

To characterize the kinetics of YM after irradiation, bladder elasticity of irradiated animals at different time points after treatment was analyzed and compared to those of healthy animals of the same age. Two months after treatment, the urothelium was the tissue component which mostly responded to irradiation, as it is evidenced both when comparing the histogram distribution of the data (Fig. 2c), as well as the fold change of the mean respect to controls (Fig. 2d). The lamina propria showed a slightly narrower distribution as compared with controls of the same age (Fig. 2c), along with an increased collagen deposition respect to control animals (Fig. 2f). The muscle layer did not show differences at this time point.

Four months after the irradiation treatment the urothelium exhibited a large spread of moduli comparable to the control of the same age (Fig. 2c). In contrast, both lamina propria and muscle layers appeared to be particularly responsive to the fibrotic process: indeed, both appeared stiffer and more narrowly distributed than their healthy counterparts (Fig. 2c). Similarly, higher mean values were observed compared to the previous time point (Fig. 2e). When comparing mean of the median values of the distributions to healthy counterparts, the lamina propria and muscle were stiffer (Fig. 2d), stiffening that associated with an increased collagen deposition on the fibrotic bladders (Fig. 2f).

When comparing YM profile from Month 4 to Month 6, the distributions did not shift significantly towards higher YM values (Fig. 2c), suggesting that the damage remains stable over time and does not further cause stiffening at the later time point (Fig. 2e). Indeed, due to the bladder physiological aging that results in stiffening, differences on irradiated and healthy animals were less drastic in older animals, as the treatment effect was likely masked by the aging of the animals and stiffening was only reported for the urothelium (Fig. 2d). Collagen quantification in the tissue from irradiated animals showed an increase in collagen deposition compared to non-treated animals at Month 2 and Month 4,

but not at Month 6 (Fig. 2f). Nevertheless, older treated animals exhibited a less broad distribution, suggesting that the decrease in stiffness heterogeneity was caused by the replacement by collagen of tissue components due to a fibrotic response.

Interestingly, we observed that the micromechanical profile of one of the irradiated animals did not show any effect in terms of stiffening, and it was comparable to those of the control healthy animals of same age (4 months after treatment) (Fig. 3a). After following histological analysis and fibrosis evaluation, we confirmed that this specific animal had not developed bladder fibrosis (Fig. 3b), therefore highlighting the potential of micro-indentation to detect fibrosis caused by irradiation, and showing that stiffening was not detected by microindentation in the absence of histological fibrotic ECM deposition.

### Microscale mechanics in the orthotopic bladder cancer model and during disease progression.
In order to study the effect of tumor development and progression on bladder mechanics, the evolution of bladder elasticity was studied on an orthotopic rat model, in which rats were watered with water containing the bladder-specific carcinogen nitrosamine (BBN) (Fig. 4a). This model allows to monitor all stages of bladder cancer development and progression, thus mimicking the pathological processes that happen in humans.

After 2 months of BBN treatment, bladder tissues presented low-grade dysplasia, which means there are epithelial cells with the abnormal organization (Fig. 4b). The presence of abnormal cells was limited to the urothelium, which increased its thickness and cellularity, without invading the lamina propria below. Furthermore, the BBN activates inflammation pathways in the bladder. At this stage, dysplastic bladder tissues were characterized by elasticity alteration of the urothelium, which correlated with broadening of the cell layer and the presence of abnormal cells (Fig. 4b, e). The lamina propria and muscle layers exhibited YM profiles equivalent to those of control animals of the same age, and the overall tissue did not differ significantly from healthy bladder tissue.

After 4 months of BBN treatment, the bladder tissues exhibited non-invasive tumors localized in the urothelium (pTa), and there were few points of focal invasion in which few groups of cells invaded superficially the lamina propria below (pT1) (Fig. 4c). These bladder tissues showed a softening in the urothelium compared to controls (Fig. 4f) of 0.81 fold change (Fig. 4h), as well as in the lamina propria which exhibited a very broad stiffness distribution (0.78 fold change). It could be appreciated that the elasticity distribution of the muscle slightly started to shift towards lower YM values in comparison with the healthy muscle, with a fold change of 0.87 respect to control muscle tissue, even though cancer cells did not invade this tissue layer yet. YM distribution of the whole tissue was characterized by a bimodal profile: one lower peak which corresponded to the presence of tumor cells in the urothelium and in the lamina propria, and a stiffer peak that corresponded to the muscle tissue layer without the presence of tumor cells, which partially overlapped with the healthy tissue (Fig. 4f).

After 6 months of BBN treatment tumor cells were present in the lamina propria (pT1) (Fig. 4d). Also, at this time point the treated bladders were characterized by elasticity distributions shifter towards softer values for urothelium and lamina propria (Fig. 4g). The muscle tissue exhibited a bimodal distribution with one peak shifted towards lower values and a second peak that overlapped with those of healthy bladder (Fig. 4g). Likely due to the physiological aging of the animals leading to increase in tissue elasticity, fold change respect to controls were less dramatic, being the lamina 0.93 softer and muscle 0.90 softer (Fig. 4h). We

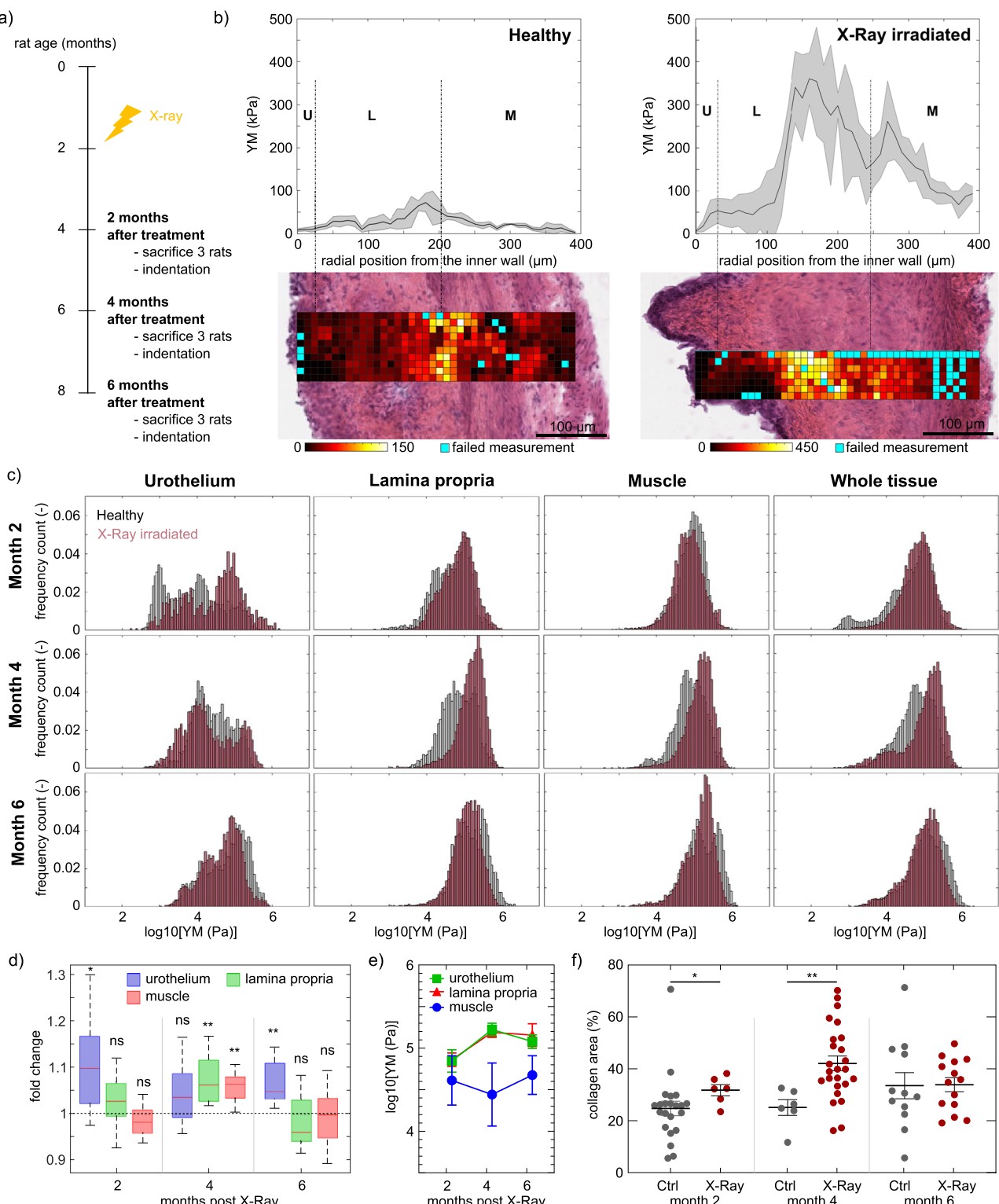

hypothesized that the main contributor to this bladder softening was the presence of tumor cells within the tissue, which are widely known to be softer than their healthy counterparts. Nevertheless, softening of the different layers was detected prior to tumor cell invasion (for lamina and muscle at month 4, and for muscle at month 6). Overall, the neoplastic microenvironment was softer than those of healthy animals, even when such softening effect was combined with the physiological aging of these animals.

**A preliminary mechanical fingerprint of high-grade human urothelial carcinoma.** Once we studied bladder tumor mechanics on murine model that allows for closer follow-up of earlier-stage tumors, we characterized the micromechanical profile of a high-grade MIBC from a patient that underwent radical cystectomy. In this context, we investigated paired surgical samples including the muscle tissue infiltrated by neoplastic cells and the non-invaded muscle tissue. This strategy allows to characterize neoplastic vs non-neoplastic tissue, but avoiding variability between donors.

**Fig. 2 Micromechanics of murine bladder in a model of actinic cystitis (X ray radiation). a** Schematic representation of the experiment: X-ray radiotherapy is used to induce actinic cystitis on the bladder. Rats are sacrificed at different time points ant tissue elasticity is assessed. **b** Representative bladder wall stiffness gradient collected with the nanoindenter at month 4: X-ray causes a stiffening of the whole bladder wall compared to non-treated healthy animals. Mechanical spatial differences within the fibrotic bladder are maintained and associated to the different tissue layers (U: urothelium, L: lamina propria, M: muscle). **c** Kinetics of YM from X-ray radiated bladders at different time points (red) and comparison to healthy animals of the same age (grey) measured both with nanoindenter and AFM. N = 3 rats per time point and condition, each tissue layer of each single rat was characterized by over 3000 force curve measurements. Data shown are logarithm in base 10 of YM's (in Pa). **d** Fold of change of mean of the median log10 YM values of tissue layers of treated rats respect to age-matched control rats. T-test showed statistically significant stiffening with respect to the control animals (mean ± standard deviation with propagated error are shown. ns=not significant, *=p value < 0.05, **=p value < 0.005). **e** Mean of the median log10 YM values of tissue layers of each rat ± SEM. 2-way Anova showed no statistical significant differences in kinetics of fibrosis development. **f)** Quantification of the bladder area that expressed collagen from control rats and X ray-irradiated animals at different time points; each symbol represents the measurement in a tissue slice, with multiple slices measured for each bladder, in three animals. Source data behind graphs and charts can be found in https://figshare.com/projects/Micro-mechanical_fingerprints_of_the_rat_bladder_change_in_actinic_cystitis_and_tumor_presence/158222.

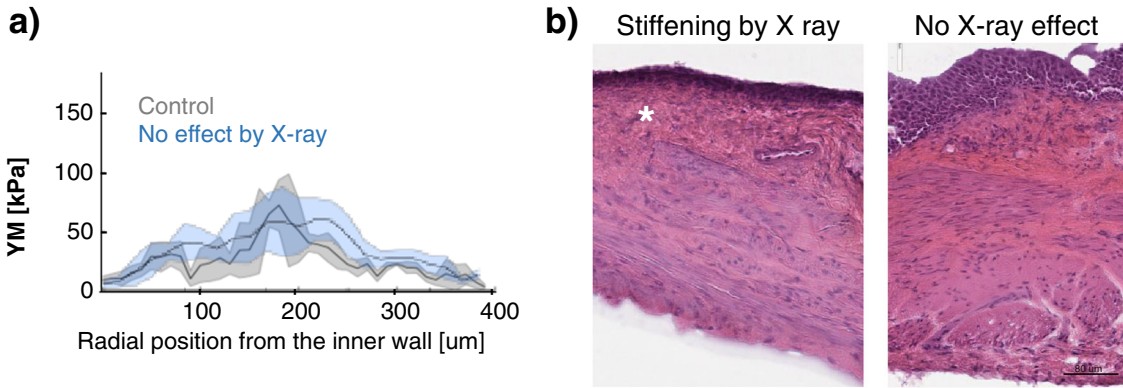

**Fig. 3 Animal with no effect by X-ray treatment. a** The micromechanical profile (blue) was comparable to those of control healthy tissues of the same age (4 months after irradiation) (grey). **b** The histological analysis showed denser deposition of ECM (*) in an X-ray-responding animal (left) accompanied by increasing of stiffness. Histology on the bladder with no stiffening (right) revealed absence of fibrotic damage (4 months after irradiation). Source data behind graphs and charts can be found in https://figshare.com/projects/Micro-mechanical_fingerprints_of_the_rat_bladder_change_in_actinic_cystitis_and_tumor_presence/158222.

Normal muscle tissue was characterized by a YM that followed approximately a Gaussian distribution after logarithmic transformation, with median value of 3.52 Log$_{10}$(YM/Pa) (33 kPa in linear scale), while neoplastic infiltration of tumor cells on the muscle tissue resulted in an increase of the mechanical heterogeneity and an overall softening of the tissue (median value of 3.28 Log$_{10}$(YM/Pa), 2 kPa in linear scale) (Fig. 5).

## Discussion

This study details the mechanics of the three tissue layers of bladder at microscale level, thus unveiling the heterogeneity in terms of elasticity of the three tissue layers, tissue stiffening for actinic cystitis and softening of all three layers during the onset and progression of bladder cancer. By providing mechanics characterization at the microscale level, here we provided novel information regarding changes in the neoplastic environment according to the way of invasion of bladder cancer that is by single cells or small cluster of cells.

To the best of our knowledge, in this study we performed the first spatial and temporal micro-mechanical mapping of the whole bladder wall, with micron-level spatial resolution, both in health condition, actinic cystitis and bladder cancer. The methodology here used was first AFM, the gold standard in mechanobiology to investigate cell mechanics. Nevertheless, the investigation of the mechanical properties of tissues needs to deal with bigger sampling regions, which means increased testing scale, surface roughness and mechanical heterogeneity of the sample. This requires first, increased Z piezo range, and second, closed indentation loop to measure all mechanical heterogeneities

(to probe very soft and very stiff regions within a big sampling area). On the other hand, nanoindentation-based mechanics investigation has been previously reported to suffer from replication issues, which has been overcome at the cell scale[24], but not yet at the tissue scale. Thus, aiming to overcome the technical challenges of testing such complex samples and increase the robustness and reproducibility of our results, as well as its eventual translation to the clinics, we here combined two indentation-based instruments: AFM and a nanoindenter, which overcomes such technicalities and allowed for validation of our own data.

We demonstrated that the bladder wall is a highly mechanical heterogeneous tissue, characterized by a gradient of stiffness from urothelium to lamina propria and muscle layer. We also showed an effect of aging on tissue mechanics, which is known to alter murine bladder histology[25]. Understanding tissue mechanics and the distribution corresponding to different anatomical tissue layers - as it was previously reported for organs as the cornea[26] or the skin[27] - is also of relevance for the bladder, as its physiological function comprehends high elastic tension and mechanical stress, and it may be crucial for bladder reconstruction purposes.

Fibrosis results from the pathological accumulation of ECM proteins due to chronic inflammation, resulting in scarring and thickening of tissue and, finally, in organ failure[15]. We used X-ray irradiation to establish an animal model of actinic cystitis, in order to characterize the mechanical fingerprints of fibrotic bladder tissue. By using the histogram representation of the YM we identified stiffening of the urothelium 2 months after irradiation, suggesting that the main effect of irradiation was on the

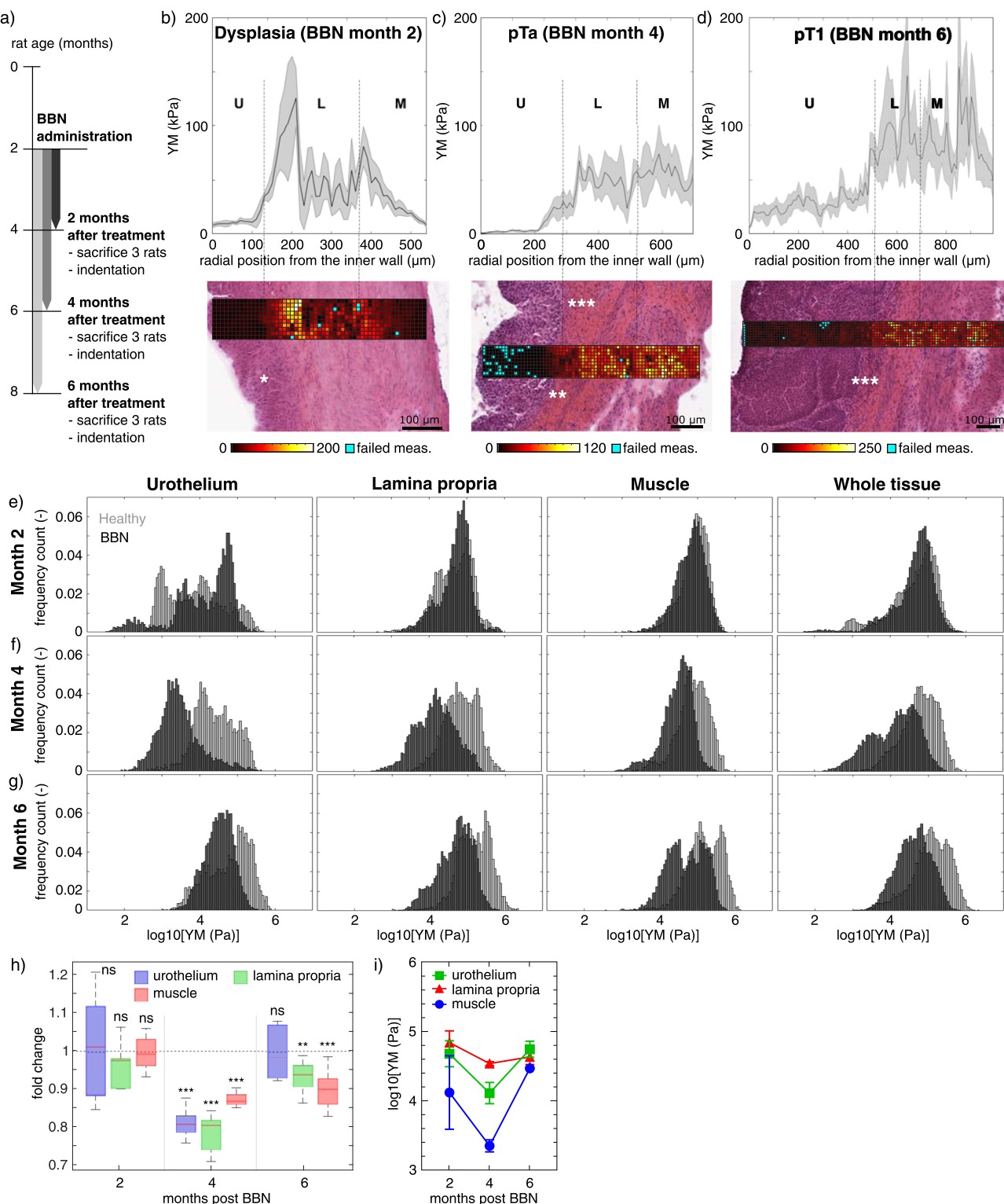

cells, which it is well known to induce cell apoptosis[28], and apoptotic cells have been previously reported to be stiffer[29]. At the second studied time point, i.e., 4 months, stiffening of the lamina propria and muscle occurred, in agreement with a previous study[30]. The YM in 6-months irradiated animal did not change compared to the 4-months treated animal, while the mechanics of bladder in control animal increased from 4 to 6 months of age, masking the effect of the treatment.

The investigation of tissue elasticity by microindentation was also able to detect the absence of bladder fibrosis on an irradiated

animal that did not develop actinic cystitis. This information supports the diagnostic potential of quantifying tissue elasticity at the microscale, which could be used to have more detailed information about the reactive stroma characteristic of fibrotic processes, which is usually investigated by histological techniques.

We also observed an increase in tissue compliance in the presence of bladder tumor, both in the rat model and in humans, similar to what has been reported for liver cancer tissues[31]. In principle this observation may seem controversial and opposite to the general tendency of solid tumors (i.e., breast[32] and lung[33]

**Fig. 4 Micromechanics of murine bladder in a model of bladder cancer (BBN). a**) Animal model establishment. Representative bladder wall stiffness gradients collected with nanoindenter at (**b**) 2 months of BBN treatment, where urothelium dysplasia is marked by *; (**c**) 4 months of BBN treatment, where pTa tumor limited to the urothelium without invading the lamina propria is marked by **. pT1 tumors in which urothelial tumor cells break the basal membrane and invade the lamina propria below at 4 and (**d**) 6 months BBN treatment are marked by ***. U: urothelium, L: lamina propria, M: muscle. YM's from BBN treated bladders (black) at (**e**) 2 months, (**f**) 4 months and (**g**) 6 months of BBN treatment; and comparison to healthy animals of the same age (grey) measured both with nanoindenter and AFM. N = 3 rats per time point and condition, each tissue layer of each single rat is characterized by over 3000 force curve measurements. Data shown are logarithm in base 10 of YM's (in Pa). **h** Fold of change of mean of the median log10 YM values of treated rats ± standard deviation (with propagated error) respect to control rats. T-test showed statistically significant softening with respect to the control animals (mean ± standard deviation with propagated error are shown. ns=not significant, *=p value < 0.05, **=p value < 0.005, ***=p value < 0.0005). **i** Mean of the median values of each rat. 2-way Anova showed that statistically significant differences in the BBN model were observed in the urothelium from month 2 to month 4, and from month 4 to month 6; and for the lamina propria from month 4 to month 6. Source data behind graphs and charts can be found in https://figshare.com/projects/Micro-mechanical_fingerprints_of_the_rat_bladder_change_in_actinic_cystitis_and_tumor_presence/158222.

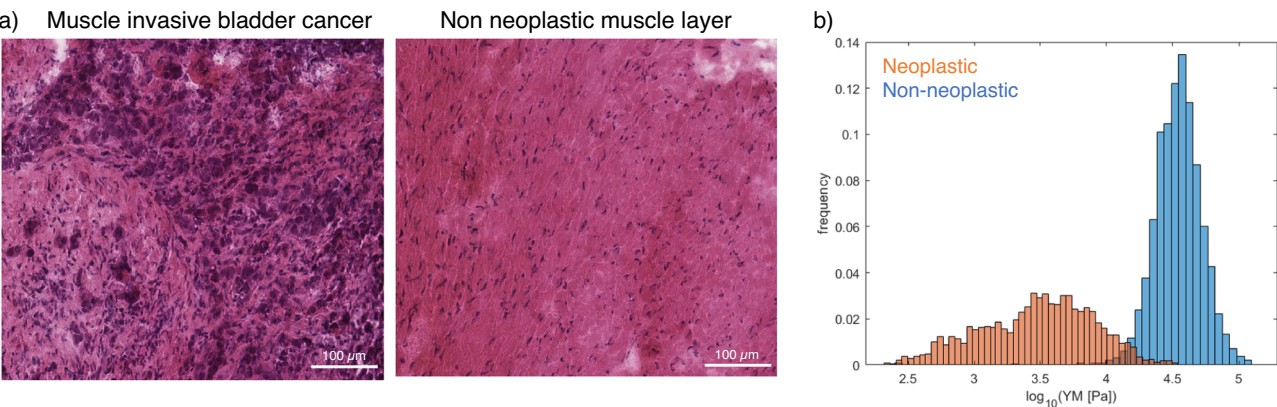

**Fig. 5 Micromechanics of high grade human bladder cancer. a** Hematoxylin-eosin staining of muscle invasive bladder cancer and paired non-neoplastic muscle tissue from a patient with high-grade urothelial carcinoma. **b** Mechanical profile of neoplastic (orange) and non-neoplastic (blue) muscle tissue from the same patient (n = 1). Data collected with the nanoindenter. Data are shown as logarithm in base 10 of YM's (in Pa). Source data behind graphs and charts can be found in https://figshare.com/projects/Micro-mechanical_fingerprints_of_the_rat_bladder_change_in_actinic_cystitis_and_tumor_presence/158222.

cancer, dominated by collective cell invasion[34]), where stiffening usually takes place. Tissue stiffening is also observed when studying urological organs as the prostate or the testis by non-invasive macroscale techniques[4]. Nevertheless, by providing spatially resolved micro-mechanical maps, we here were able to measure the contribution to tissue elasticity both from cells and ECM. In the MIBC tumor we found an increase in the heterogeneity of the elasticity of the tissue when the tissue is invaded by neoplastic cells, as previously reported for breast cancer[32]. In addition, it has been previously established that tumor cells are softer than their benign counterparts, and that softening of cancer cells increases with increased malignancy[35].

On the other hand, ECM stiffening is increasingly recognized as a major mechanical signal, which alters cell behaviour and in part confers cancer cell hallmark capabilities including sustained growth, invasion, and metastasis[36]. Such increase in ECM stiffness is mainly associated to increased collagen deposition[37,38] especially at the invasive front of tumors, which furthermore often corresponds to the stiffest region of the tumoral tissue[39]. One example of tumoral fibrotic stroma is the case of breast cancer, in which breast tumors are characterized by increased collagen deposition together with increased linearization and thickening of collagen fibers[40,41]. Those tumors characterized by fibrotic stroma deposition are known to be stiffer, as breast and pancreas[42]. In addition, it has been reported for NMIBC patients an association with COL1A1 and COL1A2[43]. Therefore, it would be eventually interesting to study decellularized bladder tissues to investigate the contribution of ECM to the whole organ stiffness, and study eventual associations with the increase in collagen

expression reported in non-muscle invasive bladder cancer patients with poor prognosis[43].

Furthermore, in bladder tumors, cancer cells migrate as single cells or small nests[33], contrary to tumors in which stiffening is reported, e.g., breast, prostate and lung cancer[33]. Single-cell invasion, can be divided into mesenchymal and amoeboid migration[33]. Few studies have shown that muscle-invasive bladder cancer promotes enhanced contractility of cells using amoeboid migration, which contrary to mesenchymal migration, takes place when surrounding matrix is relatively soft[44]. By investigating elasticity profiles of bladder tumor in the rat model, we detected softening in those tissue layers under the tumor when the tumor did not invade yet. These results indicate that the tissue layers undergo mechanical remodelling being primed before it is invaded by the tumor cells, thus highlighting the clinical potential of measuring tissue elasticity by microindentation for early bladder cancer prognosis. This observation goes in line with a previous study, which reports that mechanical remodelling of tissue surrounding neoplastic cells precedes invasion in head and neck squamous cell carcinoma spheroids[45]. Further tests on clinical specimens are needed to validate the information here obtained by investigating tissue mechanics using a preclinical model.

A limitation of our study is that bladder specimens were snap-frozen prior to mechanical measurements. Even though there was no chemical treatment performed and samples were considered fresh/frozen, it cannot be excluded that the freezing/unfreezing procedure might include some artifact on the specimens, raising the possibility of selection biases. Aiming to reduce this effect we

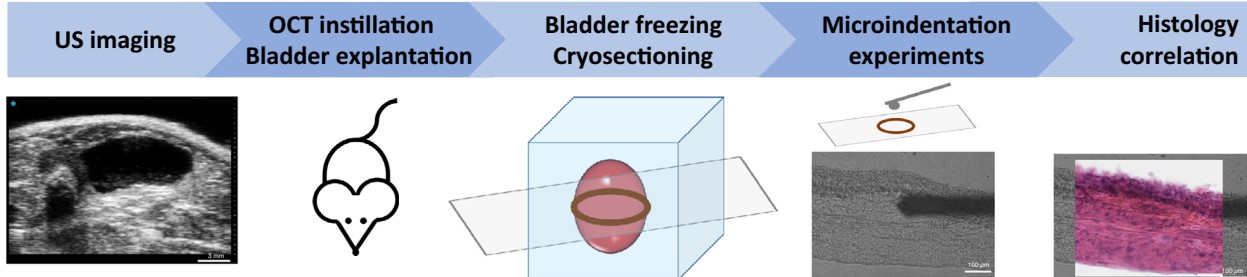

**Fig. 6 Methodological approach for bladder stiffness characterization.** Sample preparation protocol and pipeline of experiments. First, ultrasound (US) imaging is performed in order to determine the physiological OCT cryoprotectant bladder instillation volume. Animals are then sacrificed, the cryoprotectant is instilled into the bladder through a catheter and the bladder is then explanted and frozen. Cryosections are prepared and microindentation experiments are performed on the tissue slides both by AFM and nanoindenter. Afterwards, histology analysis was performed to confirm the location of the bladder anatomical layers.

used a cryoprotectant (OCT) that protects the sample from ice crystal formation and used a snap freezing protocol. Furthermore, we here performed a comparative study between different conditions, therefore the samples were always subjected to identical protocol. Previous studies have shown little impact on the mechanical properties on freezing tissue[46]. However, working with frozen tissues has several advantages, and different AFM studies used fresh-frozen clinical specimens[27,47–50]. Furthermore, frozen tissue allows to prepare semithin sections, which allows histological analysis and correlation with histological examination.

Another limitation of our work has to do with the mechanical modeling: when estimating the elastic properties, we neglected viscous effects. We assumed the values we report can be considered characteristic of a low-frequency material response, where viscous effects play a negligible role. Nevertheless, the Young's Modulus only gives a partial insight into the mechanical response of the material. For example, a recent work[51] highlighted how different amounts of collagen affect both tissue stiffness and viscosity in cellular aggregates. Future studies should focus on the characterization of layer-specific viscoelastic effects.

This study highlights the intrinsic mechanical heterogeneity of the different bladder layers. By providing high-resolution micromechanical maps, which account for the three anatomical layers of bladder, we report an alteration of the mechanical properties of bladder tissue in the pathological conditions of actinic cystitis and tumor. Such mechanical fingerprints could eventually pave the way for future clinical diagnostic and prognostic tools, likewise laying out eventual hints for bladder reconstruction purposes.

## Methods

**Ultrasound image analysis.** Fischer female rats were handled according to the ethical protocol #1114. In order to induce a physiological stretching stress of the bladder, ultrasound (US) imaging was performed prior to sacrificing the animals in order to infer the instillation volume that corresponds to a physiologically stretched bladder (Fig. 6 and Supplementary Fig. 3). US imaging was performed on a Vevo3100 LAZR-X Imaging Station equipped with a MX250D transducer (FUJI-FILM VisualSonics, Amsterdam, the Netherlands). US signal was collected acquiring axial sections of the rat bladder using the following settings: frequency: 21 MHz; gain: 15 dB; step size: 200 µm. B-mode 3D ultrasound images of the rat bladder were acquired and analysed with VevoLab 3.2.5 software.

**Rat model of actinic cystitis.** Two months old, 150–175 grams female Fisher rats (Charles River, Germany) were housed in the animal facility at IRCCS San Raffaele Hospital under standard conditions (temperature: 22 °C ± 2; humidity: 50 ± 10%; light/dark cycle: 12-h light and 12-h dark). After a 1-week period of acclimatization, the rats were x-ray irradiated. Animals were anesthetized with isofluorane 2–4% 0.3–0.8 L, and the bladder filled with 450 µl of sterile saline solution through PE50 catheter[52]. The bladder was irradiated with a single show of 20 Gy. The radiation dose was delivered using a dedicated small animal micro-irradiator (X-RAD225Cx SmART, PXI North Branford, CT, USA) with micro-cone beam

computed tomography (CBCT) guidance. The anesthetized rats were positioned prone on the animal stage and CBCT images were acquired using the following settings: tube voltage = 40 kVp, current = 5 mA, voxel size = 0.2 mm³. The bladder was contoured on the CT scan and three equal-sized dose beams were set at 130°, 180° and 230° angles respectively, using a collimator of 10 × 10 mm². Dose distribution was calculated by means of a Monte Carlo algorithm[53] and the mean dose to the bladder was adjusted to the prescribed dose of 20 Gy. Irradiation settings were: tube voltage = 225 kVp, current = 13 mA. Delivery time ranged approximately between 2 and 5 min/field and the entire procedure (CT imaging and radiotherapy) was performed within 20–25 min/animal. Rats were then sacrificed 2, 4 and 6 months after radiotherapy, (3 rats per condition) and bladders were prepared for histology and mechanical testing.

**Rat model of bladder carcinogenesis (BBN model).** Two months old female Fisher rats (Charles River, Germany) were housed in the animal facility at IRCCS San Raffaele Hospital under standard conditions (temperature: 22 °C ± 2; humidity: 50 ± 10%; light/dark cycle: 12-h light and 12-h dark). After a 1-week period of acclimatization, the rats were evenly divided in two groups: one group (tumor) that was watered with 0.05% N-(4-hydroxybutyl)nitrosamine (BBN; Sigma Aldrich) and the second group (control) watered with normal water. Rats were then sacrificed 2, 4 and 6 months after treatment initiation, (3 rats per condition) and bladders were prepared for histology and mechanical testing.

All procedures and studies involving animals were performed under protocols approved by the IRCCS Ospedale San Raffaele Animal Care and Use Committee, and in accordance with national and international standard guidelines.

**Human specimens.** Paired specimens of non-neoplastic tissue and urothelial carcinoma of the bladder were obtained from one patient (male, 62 years, with MIBC-pT2 G3) through the Unit of Pathology at the IRCCS Ospedale San Raffaele (Milan, Italy). A formal written consent was obtained by the local Institutional Review Board (Ethic Committee IRCCS Ospedale San Raffaele; amended version of URBBAN protocol approved on November 2020). Data collection and all experimental protocols were approved by the Ethic Committee IRCCS Ospedale San Raffaele, in accordance with the relevant guidelines and regulations outlined in the Declaration of Helsinki. All methods were carried out in accordance with the approved guidelines. All patients signed written informed consent agreeing to supply their own anonymous information for this and future studies.

**Sample preparation for mechanical analysis.** For the rat specimens, bladder ultrasound imaging was performed in order to characterize the instillation volume that causes a physiological stretching of the bladder. Animals were then euthanized by CO2 and through bladder catheterization (22 G cannula, BD, Italy) the cryoprotectant OCT (Bio-Optica, IT) was instilled into the bladder (volume previously defined by US imaging, ranging from 200–400 µl depending on size and stretch ability to each bladder) (Fig. 6). The urethra was closed and the bladder was explanted. In order to preserve specimen integrity, bladders were snap-frozen in tissue embedding medium (OCT Compound for Cryostat Sectioning) at −80 °C (isopentane and dry ice)[54]. Human specimens were frozen in the same way.

For the mechanical analysis 50 µm thick tissue sections were prepared using a microtome cryostat, and the fresh-frozen sections were thawed at room temperature on polarized superfrost glass slides in order to immobilize them for mechanical measurements. OCT was removed with phosphate buffer solution (PBS) wash prior mechanical testing. Paired 10 µm thick frozen sections were thawed, formalin fixed and hematoxylin-eosin stained for comprehensive histological analyses.

**Indentation techniques.** The Young's modulus (YM) of bladder tissue specimens was characterized using the Chiaro nano-indenter (Optics11), and the Bioscope

**Fig. 7 Indentation techniques. AFM (left) and Chiaro nanoindenter (right).** The main difference between both indentation instruments are the z piezo range, the detection of cantilever deflection and the fixed parameter during measurements.

Catalyst AFM (Bruker). While both instruments provide similar results in terms of measured YM values, the main difference between the two techniques resides in the apparatus used to measure the force applied to the sample (Fig. 7).

**AFM-based indentation measurements**. Young's modulus values of the different layers of the bladder tissue were determined by fitting the Hertz model[55,56] to sets of force versus indentation curves (simply force curves, FCs) acquired by AFM on 50 μm thick tissue sections, as described elsewhere[24,57,58], which is accurate as long as the indentation δ is small compared to the radius R:

$$F = \frac{4}{3} \frac{E}{1 - \nu^2} R^{\frac{1}{2}} \delta^{\frac{3}{2}} \tag{1}$$

In Eq. (1), ν is the Poisson's coefficient, which is typically assumed to be equal to 0.5 for incompressible materials, and E is the Young's modulus.

Custom monolithic borosilicate glass probes consisting of spherical glass beads (SPI Supplies), with radii R in the range of 5–10 μm, were attached to tipless cantilevers (Nanosensor, TL-FM) with nominal spring constant k = 3–5 N/m. Probes were fabricated and calibrated, in terms of tip radius, according to an established custom protocol[59]. Variations in the contact radius were due to the technical availability of the single probes during the measurements and to internal variations of the glass beads (within the same batch) used for the fabrication of the colloidal probes. The spring constant of the cantilever was measured using the thermal noise calibration method[60,61] and corrected for the contribution of the added mass of the sphere[62,63]. The deflection sensitivity was calibrated in situ and noninvasively before every experiment by using the previously characterized spring constant as a reference, according to the SNAP procedure[24].

All mechanical measurements have been performed with tissue samples immersed in PBS solution. Sets of FCs (force volumes, or FVs) were collected in selected regions of interest identified exploiting the accurate alignment of optical and AFM images obtained using the Miro software module integrated in the AFM software. The optical access and the design of the tissue slices allowed to move the probe directly over the different layers of the bladder (urothelium, lamina propria, muscle layer) and to localize the regions of interest to be analyzed for local mechanical properties.

Each FV typically consisted of an array of 144–225 FCs, spatially separated by 5–10 μm, each FC containing 8192 points, with ramp length L = 6–10 μm, maximum load Fmax = 200–1500 nN, and ramp frequency f = 1 Hz. The maximum load was adjusted in order to achieve a typical maximum indentation the range was around 2–5 μm. Typical approaching speed of the probe during indentation was 12–20 μm/s.

Each tissue layer was characterized by collecting at least 15 independent FVs in different macroscopically separated regions of interest on three different tissue slices for each rat (i.e. for each different bladder organ). Tissue slides from the middle region of the bladder were selected in order to normalize for the bladder wall thickness (bladder cross section closer to the ends of the bladder have thicker muscle layer). Furthermore, in order to avoid bias due to sampling a particular area of the bladder, we randomly chose at least four locations within each investigated tissue section to sample the three tissue layers, taking as reference the four cardinal

points of the tissue section. In total, each layer has been characterized by more than 2000 FCs per single organ. Data analysis was performed as described before elsewhere[64].

After the mechanical test, tissue sections were fixed on 4 % paraformaldehyde (PFA) and hematoxylin-eosin staining was performed on the same slice in order to provide a retrospective confirmation of the anatomical location of each single region of interest measured during the nanoindentation experiment (Fig. 6).

**Nanoindenter measurements**. Mechanical maps on murine bladders were collected using a Chiaro nanoindenter (Optics11 B.V.) in at least three well-separated areas in each collected slice. Dimensions wise, a pixel size of 10 μm was chosen with an overall map dimension of 100 μm in the tangential direction (through the different layers), and whatever length was necessary to cover the whole thickness of the bladder wall, from urothelium to muscle (typically a few 100 s of μm).

Mechanical characterization of human bladder specimens was performed with the same Chiaro nanoindenter. Maps of up to 1 mm by 1 mm (pixel size of 10 μm) were collected in the three different tissue layers from two different tissue pieces per bladder. Three to five tissue slides per tissue piece were measured. The Hertz model (Eq. 1) was fit to the force curves.

The tissues were probed using cantilevers with ~0.5 N/m spring constant and ~9 μm radius. All measurements were performed in Indentation control mode (that is, in a closed loop so the indentation rate is constant throughout the loading phase), with 5 μm/s indentation rate and a target indentation of 5 μm; as for the case of AFM, this choice was also done to comply both with the parabolic indenter approximation of Hertzian theory and to avoid bottom effect given by the finite thickness of the sample[65,66].

To establish the optimum indentation depth, YM's obtained at different indentation ranges were compared, and the surface roughness was analyzed by means of optical profilometry (Veeco WYKO NT9100 and OCT, (Supplementary Fig. 4). The latter techniques showed a scale-dependent roughness, with arithmetic roughness values of around 2–300 nm when areas of ~10 μm radii were analyzed (that is, the scale of the contact of the indenter sphere). To ensure conformal tissue/indenter contact, we performed Hertzian fit after excluding the first portion of contact. We considered a fit to be valid when $R^2$ was > =0.90. We also compared the fitting results for different indentation depth ranges (1–3 μm, 3–5 μm): we observed that at different indentation depth the YM remained mostly the same, with a slight increase in the number of rejected fits with increasing depth of indentation (Supplementary Fig. 5), hinting at a mild non-linear behavior at larger strains. We chose to limit the analysis to shallow indentations (0.2–1.2 μm) to maintain more consistent results and more easily compare the results to the AFM dataset; this choice also assured that the maximum indentation was significantly larger than the typical roughness value.

**Histological analysis pairing**. Bladder cryosections of 10 μm thick were prepared, OCT was washed away with PBS and tissues slides were fixed on 4 % PFA. Then Hematoxylin Eosin (HE) staining was performed as follows; slides were washed twice on MilliQ water and cell nuclei were stained with Hematoxylin for

50 seconds, next washed for 5 min in MilliQ water and incubated on Eosin for 15 seconds. After washing, tissue slides were dehydrated on an increasing gradient of ethanol and then incubated on xylene as a clearing agent. Samples were then mounted with Eukit. A comprehensive histopathological analysis from HE slides was performed by our experienced uro-pathologist (R.L.) blindly with respect to the mechanical data. In addition, same tissue slices previously used for mechanical measurements (50 μm thick) were fixed and HE stained in order to confirm the tissue layers (20 seconds Hematoxylin and 15 seconds Eosin).

**Collagen quantification**. Collagen quantification from HE stained tissues was performed by quantifying collagen birefringence from images acquired with polarized light in dark field microscope (Zeiss AxioImager M2M). All images were captured using 10X objective (APOCHROMAT 10X - NA 0. 4 5) and analyzed with ImageJ software. Images were first transformed into 8 bit images, threshold was manually adjusted in order to remove background noise and collect all positive pixels from the tissue. The percentage of area fraction above the established threshold was quantified.

**Statistics and reproducibility**. The median YM value per tissue layer and rat was calculated. Since both AFM and nanoindenter instruments provide comparable YM values and the force curves are conceptually measured in the same way, reported results here represent data in duplicate collected both by AFM and nanoindenter, and pulled median values as obtained by both instruments per rat and tissue layer.

In order to analyze the effect of X-ray irradiation and BBN treatment in each tissue layer, we calculated the median YM fold changes with respect to the healthy rats. To do so, we first averaged each rat's AFM and nanoindenter median YM value for each tissue layer at a specific time point. This gives 3 median values (one per rat), at each of the studied conditions. Since we were interested in the (mechanical) changes as a consequence of a diseased condition, we paired treated/control animals. As every rat measured was an end-point, we assumed that any diseased condition could be originated from any of the healthy rats. This means, the possible fold-changes per conditions are the combinations of 3 control and 3 treated rats per time point;

| | |
|---|---|
| (C = control, T = treated) | Rat1$_T$/Rat1$_C$, Rat2$_T$/Rat1$_C$, Rat3$_T$/Rat1$_C$, Rat1$_T$/Rat2$_C$, Rat2$_T$/Rat2$_C$, Rat3$_T$/Rat2$_C$, Rat1$_T$/Rat3$_C$, Rat3$_T$/Rat3$_C$, Rat3$_T$/Rat3$_C$. |

This gives 9 number per treatment and time point. If we subtract 1 (no fold change from control rats) from each, we can then test the resulting distribution with a 2-way t-test to verify the hypothesis that it has a non-zero mean.

To study kinetics of treatment 2-way Anova with Tukey's multiple comparison test was performed, by comparing the mean of the median YM values of each tissue layer of each single rat at the different time points. We checked for normality of the YM distributions by performing quantile-quantile (QQ) plots per rat and tissue layer for both instruments. For those YM distributions that were not normally distributed, a non-parametric statistical test was performed (Mann-Whitney test).

**Reporting summary**. Further information on research design is available in the Nature Portfolio Reporting Summary linked to this article.

## Data availability

All source data behind graphs and charts can be found in https://figshare.com/projects/Micro-mechanical_fingerprints_of_the_rat_bladder_change_in_actinic_cystitis_and_tumor_presence/158222. The source data is included in supplementary data 1–9. Any remaining information can be obtained from the corresponding author upon reasonable request.

## Code availability

Code available from the corresponding author on reasonable request.

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

## Acknowledgements

We thank H. Holuigue and E. Lorenc for support in AFM measurements. This research was funded by the European Union's Horizon 2020 research and innovation program, under the Marie Skłodowska-Curie Action grant agreement No. 812772 (project Phys2Biomed) and received funding from the European Union's Horizon 2020 research and innovation program under grant agreement 801126 (EDIT).

## Author contributions

Conceptualization, L.M.V., A.P., M.A.; Methodology, L.M.V., M.C., A.P., M.B., E.A., I.L., F.P., C.V., R.L.; Formal analysis, L.M.V., M.C., M.B., C.V., A.P., M.A.; Investigation, L.M.V., M.C., M.B.; Resources, M.A., A.P., P.M.; Data curation, L.M.V., M.C., M.B., A.P., M.A.; Writing-original draft, L.M.V., M.A.; Writing review & editing, L.M.V., M.C., M.B., E.A., F.P., C.V., P.M., K.B., A.S., A.P., M.A.; Funding acquisition, M.A., P.M.; Supervision, M.A.

## Competing interests

MB and KB are employed at Optics11 B.V. All other authors declare no competing interests.
