## [Peer Review File · Communications Biology]

Reviewers' comments:

Reviewer #1 (Remarks to the Author):

Report on paper "Micro-mechanical fingerprints of healthy bladder and during the follow up of actinic cystitis and bladder tumor"

The authors report indentation experiments (AFM and indenter) on bladder tissue mechanical properties in combination with histological analysis and collagen quantification, in order to test the possible impact on actinic cystitis and bladder cancer invasion. They tested healthy tissue layers from the urothelium to the outside while studying the deposition of ECM. They found alterations of the tissue layers when testing pathological cases.

This is a very important study, which is very well designed, using different complementary means of investigation of the spatio-temporal layer properties and disease stage. The experiments are well analyzed using a large measurement database, plotted as histograms with good statistical analysis. The results are quite convincing and deserve to be published.

However there are a few improvements that could be done before publication. They are listed below.

Questions:

The paper is already well documented but a few more references should be considered

- Some relevant papers on ECM deposition (in particular collagen) and the possible stiffening of tumors exist and could be discussed more in details (Provenzano et al. BMC Medicine 2006, Whatcott et al. Clin. Cancer Res, 2015, Brooks et al. Oncotarget, 2016), and compared with the present results

- Elasticity effects might not be sufficient to explain the overall cell behavior. Actually authors mention elasticity and viscosity of cells and soft tissues at the beginning but do not discuss such effects any further, therefore they could mention other recent works (Abidine et al. Biophys J, 2018, Tsvirkun et al. J Biomech, 2022) precisely dealing with bladder tumor cells or spheroid viscoelastic properties when collagen is present.

Minor points

- ref 20 and 21 are not appropriate. Please use papers instead

- page 19 ...softening of the different layers was detected prior to tumor cell invasion

- page 36: Y axis on figure 3 should have "collagen area"

Reviewer #2 (Remarks to the Author):

The work of Martinez-Vidal et al. presents the mechanical characterization of rat bladder tissue, comparing normal tissue with two pathological conditions, actinic cystitis, and urothelial bladder cancer. The methodological approach consists of the use of two techniques: the atomic force microscopy (AFM), a fairly established technique in mechanobiology, and the nanoindenter. The authors can attribute the mechanical properties of the tissue to different anatomical layers and assess the differences in elasticity to diverse pathophysiological conditions.

The manuscript is well presented and technically sound, the discussion and conclusions appropriate with the data. I think this work is relevant in the field of mechanobiology and contributes to explore mechanical descriptors in clinically relevant samples. I would like to recommend it for publication in Communications Biology, assuming the following points are cleared out:

- 1- My main concern is the approaching speed for the AFM experiments ranges from 16 to 30 $\mu\text{m/s}$. If

the velocity has not been kept constant during the experiment or during the different experiments to compare the specimens, this can constitute an important source of error in the calculated elasticity. In the same line, the authors should justify why the radius of the glass beads has not been kept constant through the experiments.

2- Can the authors further discuss the statistical relevance within the tissue, organ, and different animals in view of the heterogeneity and complexity of the sample? 15 different regions on 3 different tissue slices per condition can lead to characterize an area of a particular elasticity which would not represent the organ per se?

3- In figure 2, description of the parameters is rather general. Specify the ranges used in this study, particularly for Z piezo range and indentation.

4- In the figures, the labeling could be improved to facilitate the reading (for example, labels of figure 2a), or figure 3b), axes of the scale bar...)

5- I suggest that the authors could add a paragraph discussing in more detail the interest of using two different mechanical techniques and what brings to this study the use of both in parallel, how these techniques compare and complement each other.

Reviewer #3 (Remarks to the Author):

This manuscript reports a study of the mechanical properties of sliced bladders using an atomic force microscope and a nano-indenter. The measurements showed the existence of mechanical heterogeneity in healthy bladder layers and an alteration of the mechanical properties of bladder tissue in the pathological conditions of actinic cystitis and tumor. Overall, the manuscript presents a convincing study on the relationship between the mechanical properties of bladder tissue and its pathological state. In my opinion, the work needs to undergo some changes before it is in a suitable condition for publication.

1. The authors showed the surface roughness of sliced bladder tissue, which affects the Young's modulus estimated from force curves, and indeed observed the mechanical heterogeneity in healthy bladder layers. How was the roughness different among bladder layers? How was the contact point of force curves determined in the sliced tissue with the surface roughness?

2. The authors mentioned that since both AFM and nanoindenter instruments provide comparable YM values and the force curves are conceptually measured in the same way (in healthy bladder), reported results here represent data in duplicate collected both by AFM and nanoindenter, and pulled median values as obtained by both instruments per rat and tissue layer. However, this has not been proven in other bladder tissues in the pathological conditions of actinic cystitis and tumor. Moreover, the AFM and nanoindenter would be in different specifications such as the instrument sensitivity and effective spatial resolution depending on the sensors and probe size. Thus, the data obtained in both instruments should not be plotted together in a single figure but separately in different figures, in which the quantifications are evaluated.

3. Two-way ANOVA and t-test were used to compare the measured groups. It should be checked whether the data (e.g., the median of the mechanical distribution in bladder layers) are normally distributed, or the nonparametric statistical tests should be used.

4. Supplementary Figure 3 showed the indentation depth-dependence of the measured stiffness where the scale of LUT in the shallow was different from that in the medium and deep regions. Is it true that the Young's modulus estimated from the shallow region was almost one-order magnitude higher than that estimated from the other regions?

5. The authors should revise unclear and small characters in Figures and Supplementary Figures (e.g. Fig. 3a and d and Suppl. Fig. 3.).

Below, the Reviewers' comments are reported in bold and the reply in Italic.

Reviewer #1 (Remarks to the Author):

Report on paper "Micro-mechanical fingerprints of healthy bladder and during the follow up of actinic cystitis and bladder tumor"

The authors report indentation experiments (AFM and indenter) on bladder tissue mechanical properties in combination with histological analysis and collagen quantification, in order to test the possible impact on actinic cystitis and bladder cancer invasion. They tested healthy tissue layers from the urothelium to the outside while studying the deposition of ECM. They found alterations of the tissue layers when testing pathological cases.

This is a very important study, which is very well designed, using different complementary means of investigation of the spatio-temporal layer properties and disease stage. The experiments are well analyzed using a large measurement database, plotted as histograms with good statistical analysis. The results are quite convincing and deserve to be published. However there are a few improvements that could be done before publication. They are listed below.

We thank the Reviewer for appreciating the importance of our study and well design of our study.

Questions:

The paper is already well documented but a few more references should be considered

- Some relevant papers on ECM deposition (in particular collagen) and the possible stiffening of tumors exist and could be discussed more in details (Provenzano et al. BMC Medecine 2006, Whatcott et al. Clin. Cancer Res, 2015, Brooks et al. Oncotarget, 2016), and compared with the present results

We thank the Reviewer for their comment. We discussed in more detail the relation between collagen deposition and tumor stiffening in the discussion section (page 22): "Such increase in ECM stiffness is mainly associated to increased collagen deposition^{52,53}. especially at the invasive front of tumors, which furthermore often corresponds to the stiffest region of the tumoral tissue⁵⁴. One example of tumoral fibrotic stroma is the case of breast cancer., in which breast tumors are characterized by increased collagen deposition together with increased linearization and thickening of collagen fibers^{55,56}. Those tumors characterized by fibrotic stroma deposition are known to be stiffer, as breast and pancreas⁵⁷. In addition, it has been reported for NMIBC patients an association with COL1A1 and COL1A2⁵⁸. Therefore, it would be eventually interesting to study decellularized bladder tissues to investigate the contribution of ECM to the whole organ stiffness, and study eventual associations with the increase in collagen expression reported in non-muscle invasive bladder cancer patients with poor prognosis⁵⁸.", and included the suggested references and some additional ones (new references highlighted in yellow in the manuscript, from 52-58):

52. Oudin, M. J. & Weaver, V. M. *Physical and chemical gradients in the tumor microenvironment regulate tumor cell invasion, migration, and metastasis. Cold Spring Harb. Symp. Quant. Biol.* **81**, 189–205 (2016).
53. Erler, Janine T., Weaver, V. M. *Three-dimensional context regulation of metastasis. Clin Exp Metastasis* **26**, 35–49 (2009).
54. Acerbi, I. et al. *Human breast cancer invasion and aggression correlates with ECM stiffening and immune cell infiltration. Integr. Biol. (United Kingdom)* **7**, 1120–1134 (2015).
55. Provenzano, P. P. et al. *Collagen reorganization at the tumor-stromal interface facilitates local invasion. BMC Med.* **4**, 1–15 (2006).
56. Provenzano, P. P. et al. *Collagen density promotes mammary tumor initiation and progression. BMC Med.* **6**, 1–15 (2008).
57. Whatcott, C. J. et al. *Desmoplasia in primary tumors and metastatic lesions of pancreatic cancer. Clin Cancer Res* **21**, 3561–3568 (2015).
58. Brooks, M. et al. *Positive association of collagen type I with non-muscle invasive bladder cancer progression. Oncotarget* **7**, 82609–82619 (2016).

- Elasticity effects might not be sufficient to explain the overall cell behavior. Actually authors mention elasticity and viscosity of cells and soft tissues at the beginning but do not discuss such effects any further, therefore they could mention other recent works (Abidine et al. *Biophys J*, 2018, Tsvirkun et al. *J Biomech*, 2022) precisely dealing with bladder tumor cells or spheroid viscoelastic properties when collagen is present.

We thank the Reviewer for their comment. In this study, we focused on the quasi-static mechanical characterization of spatially resolved mechanical properties of the bladder. A full viscoelastic characterization, where each indentation is a dynamic measurement, would require massive experimental times. We chose a simple approach also in the context of mechanical tests that can be realistically performed in a clinical setting. That being said, it is indeed true that the depiction of mechanical properties of our study is partial, and that viscoelasticity plays a significant role. We added a sentence clarifying this in the discussion (page 24):

*“Another limitation of our work has to do with the mechanical modeling: when estimating the elastic properties, we neglected viscous effects. We assumed the values we report can be considered characteristic of a low frequency material response, where viscous effects play a negligible role. Nevertheless, the Young’s Modulus only gives a partial insight into the mechanical response of the material. For example, a recent work (Tsvirkun et al. *J Biomech*, 2022) highlighted how different amounts of collagen affect both tissue stiffness and viscosity in cellular aggregates. Future studies should focus on the characterization of layer-specific viscoelastic effects.”*

Minor points

- ref 20 and 21 are not appropriate. Please use papers instead

We substituted these references for papers.

- page 19 ...softening of the different layers was detected prior to tumor cell invasion

We thank the Reviewer for the grammar correction and amended the main text.

- page 36: Y axis on figure 3 should have "collagen area"

We amended the figure.

Reviewer #2 (Remarks to the Author):

The work of Martinez-Vidal et al. presents the mechanical characterization of rat bladder tissue, comparing normal tissue with two pathological conditions, actinic cystitis, and urothelial bladder cancer. The methodological approach consists of the use of two techniques: the atomic force microscopy (AFM), a fairly established technique in mechanobiology, and the nanoindenter. The authors can attribute the mechanical properties of the tissue to different anatomical layers and assess the differences in elasticity to diverse pathophysiological conditions.

The manuscript is well presented and technically sound, the discussion and conclusions appropriate with the data. I think this work is relevant in the field of mechanobiology and contributes to explore mechanical descriptors in clinically relevant samples. I would like to recommend it for publication in *Communications Biology*, assuming the following points are cleared out:

We thank the Reviewer for acknowledging the relevance of our study and the interest for the mechanobiology field.

1- My main concern is the approaching speed for the AFM experiments ranges from 16 to 30 $\mu\text{m/s}$. If the velocity has not been kept constant during the experiment or during the different experiments to compare the specimens, this can constitute an important source of error in the calculated elasticity. In the same line, the authors should justify why the radius of the glass beads has not been kept constant through the experiments.

We thank the Reviewer for pointing out this topic. We realized there was a mistake in the "AFM-based indentation measurements", the actual difference in the approaching speed was between 12 to 20 $\mu\text{m/s}$. We corrected the sentence accordingly (page 10 of the manuscript). At the same time, the reviewer is right about different approaching velocities which could affect the measurement of the Young's Modulus (or Storage modulus in rheological context); in particular, cells and tissues, that are viscoelastic, store and dissipate mechanical energy depending on the rate at which the stimulus is applied.

Nevertheless, biological specimens usually manifest a storage modulus with a weak power law relation with frequency (with exponent that can vary between 0.05 and 0.3 [Alcaraz et al. Biophys. J. 84(3), 2071-2079, (2003), Rigato et al. Nat. Phys. 13, 771-775, (2017)]), and as a consequence with the approaching speed. As a matter of fact, differences in the approaching speed of orders of magnitude would be necessary to appreciate modification in the mechanical response of the tissue. We are confident that, in our case, a factor 1.67 between the approaching speed would produce a negligible effect on the characterized Young's modulus.

Small variations in the radius of the colloidal probes were due several causes: to differences in the fabrication protocols between AFM custom procedure [Indrieri et al. Rev. Sci. Instrum. 82, (2011)] and industrial processes for Chiaro probes, to actual availability of probes during the experiments (probes can get broken easily in such experiments), furthermore, small variations in the radius of the glass beads coming from the same batch are rather common. However, the fine calibration procedure for both techniques allowed us to always keep the indentation small compared to the radius of the indenter in order to not affect the mechanical

readout [Kontomaris et al. *Eur. J. Phys.* 42(2), (2021). Muller et al. *BMC Bioinformatics*, 20(1), (2019)].
 We added a sentence in the manuscript to clarify these changes in the Radii (page 9 of the manuscript).

2- Can the authors further discuss the statistical relevance within the tissue, organ, and different animals in view of the heterogeneity and complexity of the sample? 15 different regions on 3 different tissue slices per condition can lead to characterize an area of a particular elasticity which would not represent the organ per se?

We thank the Reviewer for bringing up this topic. In this study we mechanically investigated three different tissue slides per bladder. Depending on where the section is taken from, the thickness of the bladder wall will vary due to the anatomical location within the bladder: the closer to the ends of the bladder the thicker the muscle layer will be. We tested whether there was a correlation between the anatomical position of the tested section within the bladder and the Young's modulus, the results are shown in the figure below. It is possible to observe a strong overlap of YM values, coming from sections with big differences in bladder wall thickness, even though the thinner slices showed some higher values compared to the thicker ones. (Figure A).

Figure A. Tissue sections from different macroscopic locations of the bladder were collected and YM was investigated. Due to bladder anatomy, the closer to the bladder end, the thicker the bladder wall appears as more muscle tissue is present. YM vs bladder wall thickness was plotted and no correlation was observed.

Nevertheless, we decided to measure sections from the middle region of the bladder, in order to characterize tissue sections in which the muscle layer contribution is not so prominent, and facilitate mechanical characterization of the three main bladder tissue layers: urothelium, lamina propria and muscle.

Furthermore, in order to avoid bias due to sampling a particular area of the bladder, we randomly chose at least four locations within each investigated tissue section to sample the three tissue layers, taking as reference the four cardinal points of the section.

We added this methodological information on the materials and methods section, page 10: "Tissue slides from the middle region of the bladder were selected in order to normalize for the bladder wall thickness (bladder cross section closer to the ends of the bladder have thicker muscle layer). Furthermore, in order to avoid bias due to sampling a particular area of the bladder, we randomly chose at least four locations within each investigated tissue section to sample the three tissue layers, taking as reference the four cardinal points of the tissue section."

3- In figure 2, description of the parameters is rather general. Specify the ranges used in this study, particularly for Z piezo range and indentation.

We thank the Reviewer for their comment. We improved the description of the used parameter for each indentation-based instrument. We specified that for AFM the Z piezo range used was typically 10 μm , and detailed the maximum load used by AFM (0.3-1.5 μN) and the fixed indentation range by Nanoindenter (closed loop control on indentation up to 2.5 μm).

4- In the figures, the labeling could be improved to facilitate the reading (for example, labels of figure 2a), or figure 3b), axes of the scale bar...)

We thank the Reviewer for pointing out to improve the clarity of the figures of our manuscript. We worked on the figures formatting and made them more readable.

5- I suggest that the authors could add a paragraph discussing in more detail the interest of using two different mechanical techniques and what brings to this study the use of both in parallel, how these techniques compare and complement each other.

We thank the Reviewer for their suggestion. a paragraph discussing complementarity and relevance of the combined use of both techniques has been added in page 20:

“The methodology here used was first AFM, the gold standard in mechanobiology to investigate cell mechanics. Nevertheless, tissue samples mechanical testing investigation needs to deal with bigger sampling regions, which means increased testing scale, surface roughness and mechanical heterogeneity of the sample. This requires first increased Z piezo range and second closed indentation loop to reflect/collect mechanical heterogeneities (to probe very soft and very stiff regions within a big sampling area). On the other hand, nanoindentation-based mechanics investigation has been previously reported to suffer from replication issues, which has been overcome at the cell scale²⁹, but not yet at the tissue scale. Thus, aiming to overcome the technical challenges of testing such complex samples and increase the robustness and replication of our results, as well as its eventual translation to the clinics, we here combined two indentation-based instruments: AFM and a nanoindenter, which overcomes such technicalities and allowed for validation of our own data.”

Reviewer #3 (Remarks to the Author):

This manuscript reports a study of the mechanical properties of sliced bladders using an atomic force microscope and a nano-indenter. The measurements showed the existence of mechanical heterogeneity in healthy bladder layers and an alteration of the mechanical properties of bladder tissue in the pathological conditions of actinic cystitis and tumor. Overall, the manuscript presents a convincing study on the relationship between the mechanical properties of bladder tissue and its pathological state. In my opinion, the work needs to undergo some changes before it is in a suitable condition for publication.

We thank the Reviewer for appreciating how we reported the relationship between the mechanical properties of bladder tissue and its pathological state.

1. The authors showed the surface roughness of sliced bladder tissue, which affects the Young's modulus estimated from force curves, and indeed observed the mechanical heterogeneity in healthy bladder layers. How was the roughness different among bladder layers? How was the contact point of force curves determined in the sliced tissue with the surface roughness?

We thank the Reviewer for the observation. We measured the surface roughness of the slices via optical optical interferometry (supplementary figure 2) to ensure the choices of tip size and indentation depths were appropriate (i.e. the characteristic size of asperities was smaller than the contact length). The slices were obtained with a cryostat, with a single cut. Within a slice, the surface roughness appears independent of the tissue layer, and much smaller than the contact radius. This indicates that differences in estimated material properties are not due to ill-defined boundary conditions, something that is further corroborated by the analysis of elastic moduli retrieved at different depths (see supplementary figure 3). Different mechanical properties were primarily associated to the different tissue layers.

Supplementary Figure 3. *YM's at different indentation depths obtained using the nanoindenter. YM's was analyzed at shallow (1.5 μm), medium (2.5 μm) and deep (5 μm) indentation depths. With increasing indentation depth, the number of non-valid pixels (blue) increased due to bottom effect, while YM values remain equivalent.*

Regarding the definition of contact point: for the nanoindentation experiments, the approaching portion of the load data was fitted to a piecewise function, set to 0 until contact point, and set to follow the Hertz model past it. Both contact point and elastic modulus were treated as optimization variables. AFM data was fitted according to the procedure outlined by Puricelli et al. in "Nanomechanical and topographical imaging of living cells by Atomic Force Microscopy with colloidal probes", March 2015, The Review of scientific instruments 86(033705), DOI: 10.1063/1.4915896.

2. The authors mentioned that since both AFM and nanoindenter instruments provide comparable YM values and the force curves are conceptually measured in the same way (in healthy bladder), reported results here represent data in duplicate collected both by AFM and nanoindenter, and pulled median values as obtained by both instruments per rat and tissue layer. However, this has not been proven in other bladder tissues in the pathological conditions of actinic cystitis and tumor. Moreover, the AFM and nanoindenter would be in different specifications such as the instrument sensitivity and effective spatial resolution depending on the sensors and probe size. Thus, the data obtained in both instruments should not be plotted together in a single figure but separately in different figures, in which the quantifications are evaluated.

We thank the reviewer for the comment. We indeed proved the comparability of data obtained with the two instruments and therefore of their pulled median values, by checking the histograms and quantile-quantile (QQ) plots per rat and tissue layer for both instruments, also in the pathological conditions of actinic cystitis and bladder tumor (we did not show them for sake of shortness and readability in the initial manuscript). Comparison of histograms and QQ plots show that the two datasets are indeed equivalent. Therefore in this revised version we report that comparability of YM provided by both instruments was checked (page 14) and added this information in the supplementary information (Supplementary Figure 4); we decided to keep the data from the both instruments pulled together in the main text, as the tissues are conceptually measured in the same way and we prove the comparability between YM obtained by both instruments.

Supplementary Figure 4. Representation of YM data extracted by the two instruments (AFM and nanoindenter) of each single tissue layer, at month 2, extracted from healthy, X-ray irradiated and BBN rats, shown as: mean of the median values, distribution of single YM values, and quantile-quantile (QQ) plots comparing the two distribution

Statistical methods (page 13) have been corrected: “We checked for normality of the YM distributions by performing quantile-quantile (QQ) plots per rat and tissue layer for both instruments. For those YM distributions that were not normally distributed, a non-parametric statistical test was performed (Mann Whitney test).”

3. Two-way ANOVA and t-test were used to compare the measured groups. It should be checked whether the data (e.g., the median of the mechanical distribution in bladder layers) are normally distributed, or the nonparametric statistical tests should be used.

The reviewer is right, we tested for normality (in the fold changes, where t-test were performed) with Shapiro-Wilk test and we did not reject the null hypothesis of non normality with significance level of 0.05. Furthermore, we tested for normality the distributions of AFM and Chiaro, and the cumulative ones, with quantile-quantile plot. Those are the distributions from which we extracted the median values and the two-way ANOVA was performed. Whether we found non-normal distribution we performed a non parametric statistical test. The text has been modified accordingly (Materials and methods section-page 14 and caption of Figure 3e).

4. Supplementary Figure 3 showed the indentation depth-dependence of the measured stiffness where the scale of LUT in the shallow was different from that in the medium and deep regions. Is it true that the Young’s modulus estimated from the shallow region was almost one-order magnitude higher than that estimated from the other regions?

We thank the reviewer for pointing out this topic. We realized there was a mistake in the scale of the maps and we corrected it. Young’s modulus estimated from the shallow, medium and deep region was comparable and no statistically significant differences were detected, while the number of non valid pixels increased with increasing depth of indentation.

5. The authors should revise unclear and small characters in Figures and Supplementary Figures (e.g. Fig. 3a and d and Suppl. Fig. 3.).

*We thank the reviewer for suggesting to improve the clarity of the figures of our manuscript.
We worked on the figures formatting and made all of them more readable.*

REVIEWERS' COMMENTS:

Reviewer #1 (Remarks to the Author):

The authors have addressed the comments raised by the referee. I am pleased to recommend this nice work for publication.

Reviewer #2 (Remarks to the Author):

In their rebuttal letter, the authors addressed the majority of my comments and the other reviewer's comments and modified the manuscript accordingly. I do not feel any restriction to recommend the new version of the manuscript for publication in Communications Biology.

Reviewer #3 (Remarks to the Author):

My concerns were addressed, and the clarity of the revised manuscript has improved. Therefore, I recommend this manuscript for publication in Communications Biology.

Milan, January 10th 2023

Dear Karli Montague-Cardoso, PhD
Deputy Editor, Communications Biology

We would like to thank you for your email on December 21st, 2022, and for the Reviewers' constructive comments on the revision of our manuscript (COMMSBIO-22-2829A).

We are very happy to know that the last revision was appreciated and the study is now suitable for publication in Nature Communications Biology.

We have edit the manuscript to comply with Nature Communications Biology policies and formatting requirements.

All authors and I have appreciated the time you and the Reviewers have invested in improving our manuscript.

Best regards, on behalf of all authors.

Sincerely, Massimo Alfano.